# Discovering the Ancient Tomb under the Forest Using Machine Learning with Timing-Series Features of Sentinel Images: Taking Baling Mountain in Jingzhou as an Example

Yichuan Liu [1], Qingwu Hu [1],*, Shaohua Wang [1], Fengli Zou [2], Mingyao Ai [1] and Pengcheng Zhao [1]

1   School of Remote Sensing and Information Engineering, Wuhan University, No. 129, Luoyu Road, Wuhan 430079, China
2   School of Geography and Tourism, Qufu Normal University, No. 80, Yantai North Road, Rizhao 276826, China
*   Correspondence: huqw@whu.edu.cn; Tel.: +86-189-7107-0362

**Abstract:** Cultural traces under forests are one of the main problems affecting the identification of archaeological sites in densely forested areas, so it is full of challenges to discover ancient tombs buried under dense vegetation. The covered ancient tombs can be identified by studying the time-series features of the vegetation covering the ancient tombs on the multi-time series remote sensing images because the ancient tombs buried deep underground have long-term underground space structures, which affect the intrinsic properties of the surface soil so that the growth status of the covering vegetation is different from that of the vegetation in the area without ancient tombs. We first use the highly detailed DSM data to select the ancient tombs that cannot be visually distinguished on the optical images. Then, we explored and constructed the temporal features of the ancient tombs under the forest and the non-ancient tombs in the images, such as the radar timing-series features of Sentinel 1 and the multi-spectral and vegetation index timing-series features of Sentinel 2. Finally, based on these features and machine learning, we designed an automatic identification algorithm for ancient tombs under the forest. The method has been validated in Baling Mountain in Jingzhou, China. It is very feasible to automatically identify ancient tombs covered by surface vegetation by using the timing-series features of remote sensing images. Additionally, the identification of large ancient tombs or concentrated ancient tombs is more accurate, and the accuracy is improved after adding radar features. The paper concludes with a discussion of the current limitations and future directions of the method.

**Keywords:** ancient tombs identification under a forest; spectral timing-series features; remote sensing archaeology; satellite; machine learning

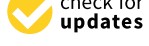



## 1. Introduction

Ancient tombs generally refer to the traces that have been left for a long time after ancient humans buried the dead, including graves, artifacts, finds, and grace yards. A large number of valuable historical relics in ancient tombs are an important part of ancient culture's splendid heritage. They are the historical epitome of various fields, such as politics and culture, in different eras, and they provide valuable and invaluable resources for various ancient sciences.

Traditional archaeological methods rely heavily on historical documents and field surveys [1]. Although the survey accuracy is high, it has flaws, such as a limited number of survey sites, high survey costs, and poor timeliness. Due to the rapid growth of space satellites and information technology, remote sensing technology has been regularly utilized in the field of archaeology [2–4]. Aerial photography archaeology led to the development of remote sensing archaeology [5]. The distribution of historical traces was initially largely determined by combining optical remote sensing satellite photos with aerial photography photographs [6,7]. Typically, the direct visual interpretation of archaeological objects is

based on the shape, size, tone, shadow, color, texture, figure, and placement of associated aspects. Large-scale archaeology in various areas, including the Mayan palace [8] and the Pyramids [9], has benefited greatly from the time-continuous remote sensing satellite photos on a world scale [10]. When combined with machine intelligence (artificial intelligence, machine learning, and deep learning) [11–13], high-resolution satellite images and UAV images with the ability to detect weak information [14,15] have made it possible for remote sensing archaeology to quickly and effectively extract various types of ancient tombs [16], such as high circular tombs in Ancient Southern Arabia [17,18], and the Frozen tombs of Iron Age Civilizations in the Altai Mountains [19]. There has to be more research performed on ancient tombs. The application of optical remote sensing technology in the field of ancient tombs is mostly based on high-precision data to detect ancient tombs with prominent features in the images. However, ancient tombs covered by forest are a significant challenge for optical remote sensing. There are ancient traces in the image that can be immediately visually interpreted, as well as indirect indicators that can be used to appraise ancient traces that are difficult to directly visually interpret, such as soil signs, vegetation signs, etc. [20,21]. They integrate remote sensing data with ground-based spectral observations of Hordeum vulgare phenological cycles to identify buried archaeological artifacts.

Optical imagery is more suitable for identifying and detecting exposed ancient surface traces than ancient traces buried in the forest. The cover of the dense jungle has greatly hampered archaeological research and development in southern China, Southeast Asia, and other locations with suitable climates and ample water resources. The LiDAR laser beam can penetrate dense vegetation to obtain precise terrain [2,22–26] and can recognize the traits implied by minute height and geographical variations [27], which provides the possibility to explore the ancient tombs covered by forest [28]. It is currently being employed to characterize the Mayan landscape's site landscape structure [29], the Angkor ruins in Cambodia [30], and the discovery of the ancient tombs of Baling Mountain in Chu's Jinan City [28,31]. However, airborne lasers currently only rely on traditional mapping algorithms [29,30] and software, making it difficult to accurately extract and express the spatial features of archaeological traces with low undulation or mixed terrain [32]. Additionally, there is no complete processing process [33]. At present, Lidar's research in the field of archaeology involves ethics and morality [34,35]. Once the data are released at will, cultural relics will be destroyed. Therefore, the conflicts between access and use, profit and sharing, international norms, and local conditions are under intense discussion. Additionally, it not only requires discussions with archaeologists but also the participation of governments and countries. In addition, the widespread adoption of LiDAR on a large or global scale is still quite challenging because it demands a lot of human, material, and financial resources and is typically only appropriate for certain places.

The main goals of this study are (1) to identify ancient tombs under the forest using digital surface model (DSM) images created by LiDAR detected in the field and easily accessible images; (2) to explore the special timing-series spectral features in Sentinel images; and (3) to use the random forest classifier in machine learning to design an automatic detection algorithm for ancient tombs under the forest based on the construction of spectral features and to generate the spatial distribution. There are still many traces in the lush jungle regions of the world. The difficulties of these jungles prevent archaeologists from using conventional field archaeology and direct picture interpretation. As a result, many of the traces in these locations are still in unknown condition. This study improves the forecast and protection of a significant number of undiscovered ancient tombs worldwide. It suggests a new research path for the challenging archaeological problem of ancient tombs under the forest.

## 2. Materials and Methods

### 2.1. Study Area

We chose Baling Mountain as a study area. It is located northwest of Jingzhou, Hubei Province, between 30°23′34.1″–30°29′18.8″N, 112°2′52.0″–112°7′40.0″E, connecting

to Jishan Mountain and belonging to the rest of Jingshan Mountain. The average altitude is 48 m, significantly higher than the surrounding areas. Jingzhou was Chu State's capital during the spring and autumn period and the Warring States period. It is one of the first 24 national historical and cultural cities announced by the State Council. Since 689 BC, when Chu State established its capital, Jinan City, 6 dynasties and 34 emperors have placed their capitals here. According to historical records, 18 Chu kings, 3 Nanping kings in the Five Dynasties, and 11 Ming kings were buried in Baling Mountain. At present, the Liaojian King Cemetery of the Ming dynasty [36], the Fengjiazhong Cemetery of Chu [37,38], the tomb of Princess Ming [39], and other ancient tombs have been excavated. The ancient tombs we studied were mainly the emperors and some royal family members. The depth of the ancient tomb is 5–20 m, generally below 5 m. Due to different times, regions, and cultures, the underground structure of ancient tombs is primarily divided into mound tombs, coffin tombs, and masonry tombs [40]. The study area is mainly the tombs of the Chu and Ming dynasty. The largest diameter of Chu's tomb is 100 m. The large tomb has not been excavated so far. The tomb of Marquis Yi of the Zeng State in this period is a mound tomb, and the area is 260 square meters. Most of the tombs in the Ming dynasty are masonry tombs. The scope of the Liao king's tomb in this period is 102 square kilometers.

Baling Mountain features densely populated ancient tombs that range in size from enormous to small, making it an ideal location for our study. The study area is shown in Figure 1. Most of the tombs are scattered throughout the mountain's peak. There are 190 confirmed ancient tombs. These serve as ground verification data alone and are not factored into the computation of the following algorithms.

### 2.2. Data Material

We mainly used DSM images, Sentinel-1 images (https://developers.google.com/earth-engine/datasets/catalog/sentinel-1, accessed on 3 October 2014), Sentinel-2 images (https://developers.google.com/earth-engine/datasets/catalog/sentinel-2, accessed on 28 March 2017), and the confirmed ancient tombs in Baling Mountain, specifically as shown in Table 1.

**Table 1.** Basic properties of research data.

| Data | Type | Resolution/m |
|---|---|---|
| DSM | Grid | 1 |
| Sentinel-1 | Grid | 5 × 20 |
| Sentinel-2 | Grid | 10/20 |
| Identified ancient tombs | Vector | – |

To further meet the needs of archaeological work in Jingzhou City, strengthen the protection of ancient sites in the city, and improve the information management of ancient sites, the Science and Technology Archaeology Research Center of Wuhan University completed the 1:2000 digital mapping work at the Great Ruins of Chu's Jinan City.

The sentinel series satellites are the core part of the Copernicus project. The Sentinel-1A is equipped with a C-band synthetic aperture radar. It has four working modes, and the basic parameters of the IW imaging method used in this paper are shown in Table 2. The all-day and all-weather imaging mission makes the Sentinel-1 data have excellent application potential and were used to monitor the growth status of crops and soil moisture [41–43].

**Table 2.** The basic parameters of the IW imaging.

| Working Modes | Width/km | Distance Resolution/m | Azimuth Resolution/m | Incidence Angle/° | Polarization Mode | Pixel/m |
|---|---|---|---|---|---|---|
| IW | 250 | 5 | 20 | 29.1~46 | HH + HH VV + VH HH, VV | 10 |

The sentinel-2A satellite carries a multispectral imager (MSI) with 13 spectral bands, covering the electromagnetic bands from visible light to short-wave infrared with a narrow channel width, mainly using 10 m and 20 m resolution bands, as shown in Table 3. Sentinel-2 combines a short revisit period, high spatial resolution, narrow bandwidth, and a large number of spectral band channels [44]. In addition, Sentinel-2 has red edge bands closely related to the physical and chemical parameters of vegetation, which can be used to describe the pigment situation and growth state of plants [45,46].

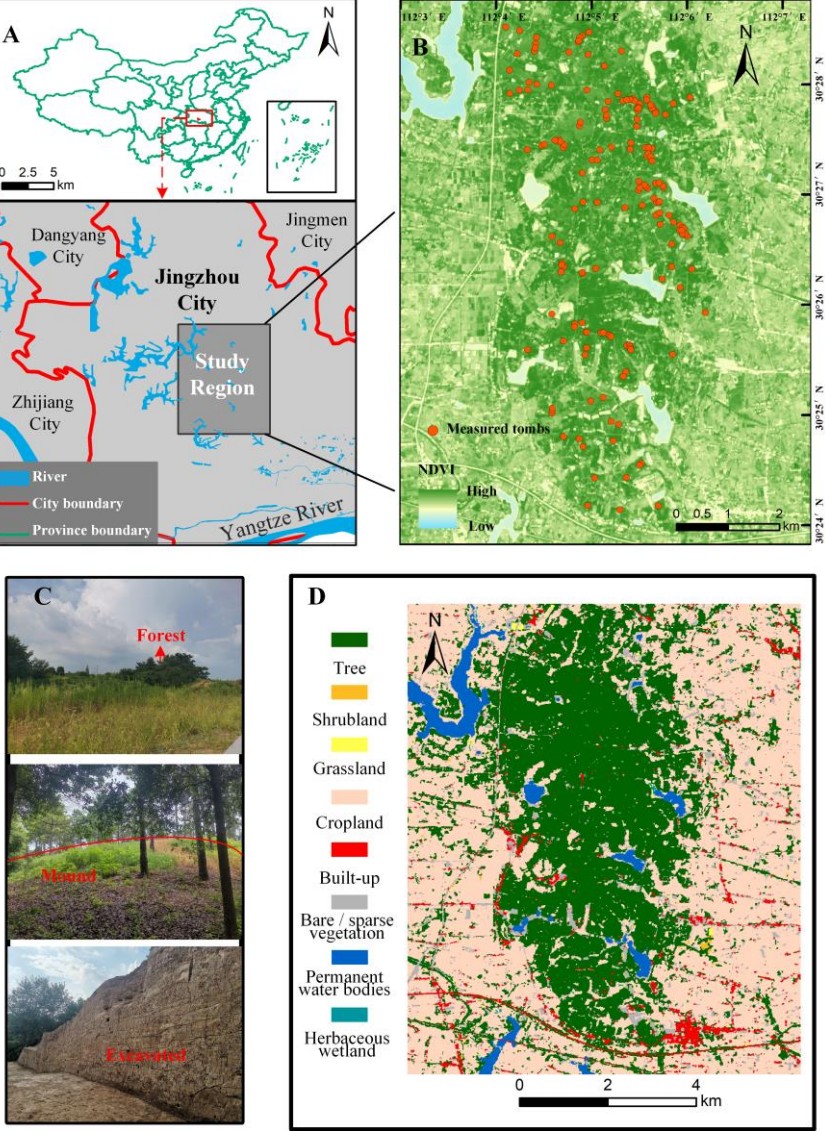

**Figure 1.** Location of the study area. (**A**) Situation of the study area in Jinzhou, and the situation of Jinzhou in China. (**B**) NDVI of study area and the distribution of confirmed ancient tombs. (**C**) Ancient tombs under the forest seen from the ground. Three pictures from top to bottom showing the ancient tombs are covered by forest, the uplifted state of the ancient tombs, and the structure of the ancient tomb. (**D**) This land cover map comes from the European Space Agency (https://developers.google.com/earth-engine/datasets/catalog/ESA_WorldCover_v200, accessed on 1 January 2020). Its name is World Cover 10 m 2020 product, and it provides a global land cover map for 2020 at 10 m resolution based on Sentinel-1 and Sentinel-2 data.

**Table 3.** Band information of Sentinel 2 image. Additionally, it excludes Band 1 and Band 9, which are not involved in the study.

| Band | Wavelength Range/nm | Spatial Resolution/m |
|---|---|---|
| B2(blue)(B) | 458~523 | 10 |
| B3(green)(G) | 543~578 | 10 |
| B4(red)(R) | 650~680 | 10 |
| B5(red edge 1) | 698~713 | 20 |
| B6(red edge 2) | 733~748 | 20 |
| B7(red edge 3) | 773~793 | 20 |
| B8(Near InfraRed) | 785~900 | 10 |
| B8A(NIR narrow 2) | 855~875 | 20 |
| B11(Short Wave InfraRed) | 1.565~1.655 | 20 |
| B12(Short Wave InfraRed) | 2.100~2.280 | 20 |

## 3. Methods

The ancient tombs under the forest—what we refer to as our research objects—were hidden by the forest and not visible in optical images. We used multi-source data to identify ancient tombs under the forest, and the overall process is shown in Figure 2. To select the ancient tombs under the forest, we first used the DSM created from the current LiDAR data in conjunction with Google Maps and land use data. Then, using the Sentinel images, we extracted the spectral features of the forest above ancient tombs, which are distinct from those without ancient tombs. To create a map of the spatial distribution of ancient tombs in the research area, we constructed an algorithm for automatically identifying ancient tombs under the forest.

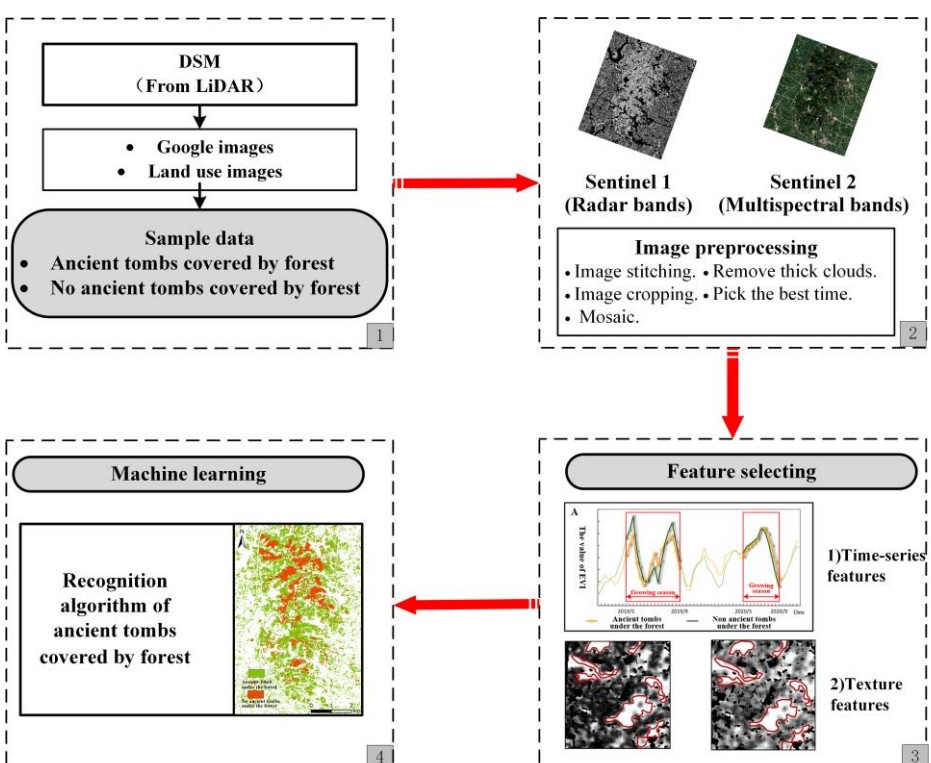

**Figure 2.** Overview of methodology.

The random forest has been widely used in remote sensing images [47,48], which shows its high prediction accuracy, good tolerance to outliers and noise, and lack of susceptibility to overfitting. However, the application of random forest is restricted to the field of remote sensing archaeology. The random forest can predict the probability of

Roman ruins at different locations [47] and detect pottery pieces [49]. However, it has not been directly applied to the study of ancient tombs.

The Google Earth Engine (GEE) platform, a free geographic cloud platform for online visual computing and analytical processing, offers many global-scale earth scientific data (particularly satellite data). The platform can simultaneously upload and retrieve its data for processing. In this study, in addition to selecting sample data of ancient tombs under the forest in Global Mapper, the preprocessing of Sentinel images and the processing of automatic detection of ancient tombs under the forest are carried out in GEE. GEE has been used in archaeology [50,51], such as in the detection of ceramic fragments [52] and archaeological mounds [53]. At the same time, it can apply machine learning methods. GEE can be seen as the perfect setting for finding ancient tombs under the forest. Additionally, GEE has the Sentinel images needed for the research, making it simple to call, process, and go on to the next stage.

### 3.1. Selecting the Sample Data of Ancient Tombs under the Forest

3.1.1. Highly Detailed DSM to Located Ancient Tombs

The ancient tombs under the forest cannot be directly identified in the images because the surface of the buried ancient tombs is covered with mounds, elevated like a hill on the surface. Highly detailed digital surface models (DSM) created by the LiDAR laser beam were used effectively in the field of archaeology [54] because they have the benefit of penetrating vegetation and showing the differences in the terrain, which can complete the selection of sample points of ancient tombs under the forest. LiDAR generates point cloud data. It can be used directly or converted into more familiar and easy-to-operate grid data, such as digital elevation models. According to the current generally accepted definition, the digital elevation model (DEM) can be seen as a general term and considered a model that only contains terrain elevation information [55]. Digital terrain model (DTM) represents bare land [56], and the digital surface model (DSM) includes the ground elevation of surface buildings, bridges, and trees, etc. DSM can express the ground undulation most realistically, so we use DSM data [57]. Additionally, the Global Mapper software is suitable for displaying subterranean historical microtopography, such as the characterization of site archaeology [58], so we use Global Mapper 22 to display the DSM images, as shown in Figure 3. From the picture, we can see the difference between the ancient tomb under the forest and the surrounding features.

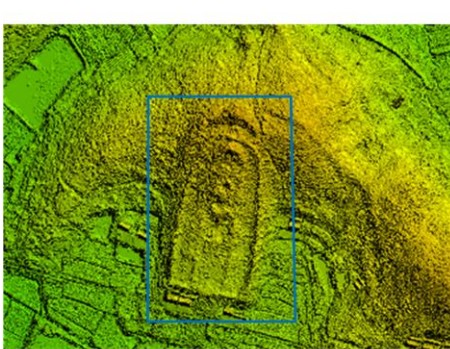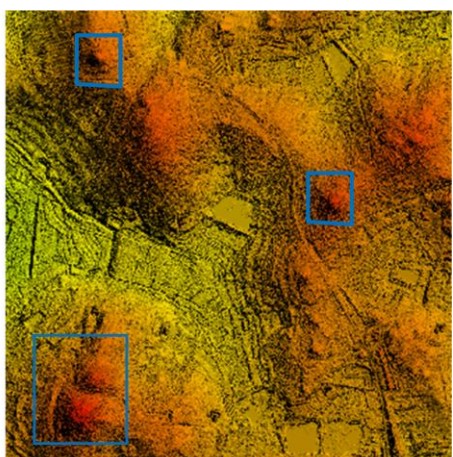

**Figure 3.** DSM display in Global Mapper 22.

3.1.2. Combined with Images to Selecting the Ancient Tombs Area under the Forest

According to the DSM, the ancient tombs showed prominent features compared with the surrounding surface, but whether they belonged to the ancient tombs under the forest needed to be visually interpreted in Google Earth and land use data. Google Images were

concentrated from June to August (summer) and November to December (winter). Most of the crops on arable land do not develop over the winter, and, in the picture, the land appeared as bare land, which may be easily distinguished from the forest. Vegetation proliferates and is stable during the summer, making it simple to distinguish it from other types of vegetation.

Since our study focused on ancient tombs under the forest, we chose two different sample types: forests with and without ancient tombs. Finally, all data types are transformed into point types, whether they contain a point or a polygon (the number of pixels included in the calculation of polygon). In the end, there are 23,314 total points, with 16,321 training points and 6993 test points used in a 7:3 ratio with the verification data. There are shown in Table 4 and Figure 4.

**Table 4.** Number of samples of ancient tombs under forest and non-ancient tombs under forest.

|  | Ancient Tombs | Non-Ancient Tombs |
|---|---|---|
| Training points | 9797 | 6230 |
| Test points | 4197 | 2670 |

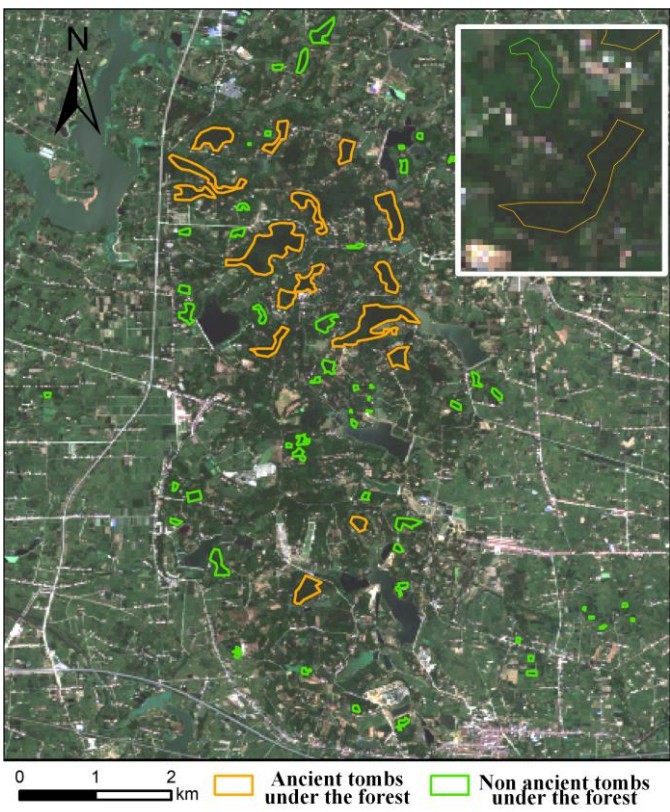

**Figure 4.** Sample data distribution of ancient tombs under the forest and non-ancient tombs under the forest.

### 3.2. Selecting Features of Ancient Tombs under the Forest in Sentinel Image

Although high-resolution images have higher accuracy, we cannot easily obtain them. Therefore, open-source remote sensing images are vital for various types of research. Among many free remote sensing images, we use the Sentinel satellite images for study.

They are mainly located at a certain depth below the underground surface. Ancient tombs are typically found in underground rooms that have been extensively excavated and physically built throughout time. The ancient objects, such as coffins, are stored in this space [59–61], forming an underground impermeable surface. Due to the tomb on the surface, it is raised like a hill on the surface. The profile of the ancient tomb from the

ground to the underground is shown in Figure 5. If there are graves under the surface, it will affect the water permeability and nutrient conditions of the soil. In the long-term atmosphere–soil–water cycle process, it will affect the growth of the long-term forest cover on it [62]. Different spectrum combinations can be used to distinguish between various states of the surface under vegetation, since the range of vegetation is thought to be a complex mixed reaction of vegetation, soil conditions, environment, and humidity [63]. Therefore, it is crucial to extract the features of spectral bands in ancient tombs under the forest and non-ancient tombs under the forest.

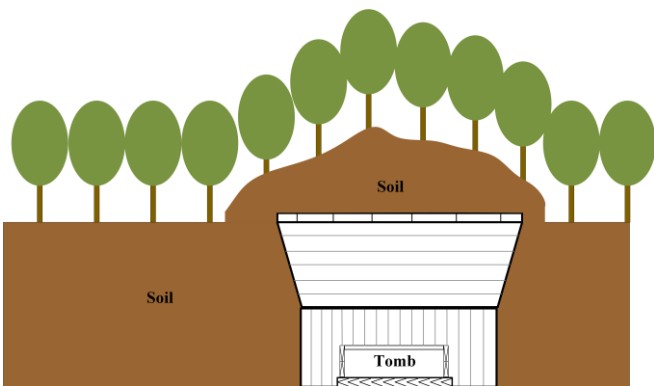

**Figure 5.** The profile of the ancient tomb from the ground to the underground.

At present, many studies have shown the differences in vegetation index [64], soil and shadow marks can indirectly identify the historical underground structure, and the best time can be determined by the intensity of the texture and tone differences of these marks on the periodic timing-series images [65]. Additionally, the radar band provided by Sentinel-1 and the vegetation index formed by the combination of Sentinel-2 multispectral bands have been used in archaeology. For example, the palaeohydrology in Foggia province (South of Italy) was used by Sentinel-1 multi-time series images suitable for distinguishing moisture and roughness [66]. The Angkor Wat and EL Mirador sites constructed temporal and spatial diversity of multi-temporal Sentinel-1 images to detect sites occluded by forest canopy [67]. In Cyprus, the covered whole wheat phenological cycle features were used to evaluate the results of multiple vegetation indices to detect buried sites [68].

### 3.2.1. Texture Features

The gray level co-occurrence matrix (GLCM) describes texture by studying the spatial correlation properties of the gray level [69]. The textural differences were examined in the near-infrared band because healthy vegetation creates reflection peaks there [70]. The angular second moment (ASM), contrast (CON), correlation (CORR), entropy (ENT), and inverse difference moment (IDM) can all be calculated by GEE for GLCM texture features in the near-infrared range. The texture features of ancient tombs under the forest is shown in Figure 6.

The ASM of the ancient tombs area under the forest is larger than non-ancient tombs area under the forest, indicating that the texture of the ancient tombs area under the forest is thicker. The CON reflects the brightness contrast between a certain pixel value and its surrounding pixel values, indicating the sharpness of the image and the depth of the texture grooves. From Figure 6, the CON of the ancient tomb area under the forest is low, and the groove is deep. The IDM reflects the image texture's homogeneity and measures the image texture's local change. The IDM of the forest buried with ancient tombs is large, indicating a lack of transitions between different texture regions and local uniformity. There are apparent differences in spectral texture features between ancient tomb areas under the forest and non-ancient tombs area, which can be used as features for separation.

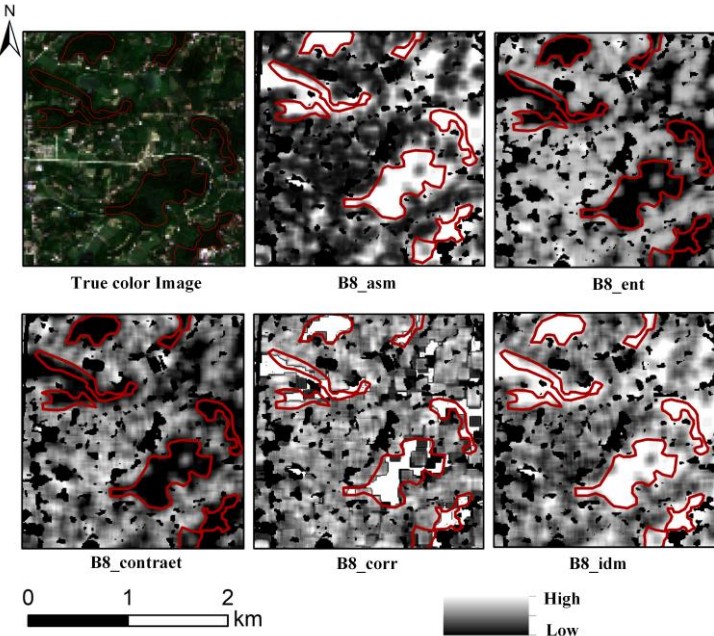

**Figure 6.** Texture representation of ancient tombs under the forest.

### 3.2.2. Timing-Series Features

We used the radar bands of Sentinel-1 and the vegetation index formed by the multi-spectral bands of Sentinel-2 to study ancient tombs. We construct timing-features as indirect markers for identifying understory remains, from which we select the best study period.

(1) Vegetation Index. The ancient tombs we studied are under the forest, so the vegetation index is very valuable for exploring the features of ancient tombs under the forest. The normalized difference vegetation index (NDVI) [71] can reflect the growth state and spatial distribution density of vegetation. Still, there is the expansion in the low vegetation coverage area and compression in the high vegetation area [72]. The calculation formula is shown in (1). The enhanced vegetation index (EVI) can improve the problems, as mentioned above, of NDVI and has higher sensitivity to detect vegetation changes [73]. The calculation formula is shown in (2). The soil brightness adjustment vegetation index (SAVI) [74] is closely related to the soil adjustment coefficient *L*. L has a value range of 0 to 1. When *L* is 0, it means that the vegetation coverage is zero; when *L* is 1, it means that the influence of soil background is 0; that means the vegetation coverage is very high and the effect of soil background is zero. The calculation formula is shown in (3). However, since the study area includes land types, such as water and buildings, the normalized water index (NDWI) [75] of water is usually greater than 0, and the normalized building index (NDBI) [76] of buildings and bare land is also greater than 0. The calculation formulas are (4) and (5). The land use types, such as water bodies, cultivated land, buildings, etc., are masked through the NDWI and NDBI indices and land use type data.

$$\text{NDVI} = \frac{\rho_{NIR} - \rho_R}{\rho_{NIR} + \rho_R} \tag{1}$$

$$\text{EVI} = 2.5 * \frac{(\rho_{NIR} - \rho_B)}{\rho_{NIR} + 6 * \rho_R - 7.5 * \rho_B + 1} \tag{2}$$

$$\text{SAVI} = (1 + L) * \frac{(\rho_{NIR} - \rho_B)}{\rho_{NIR} + \rho_R + L} \tag{3}$$

$$\text{NDBI} = \frac{\rho_{SWIR1} - \rho_{NIR}}{\rho_{SWIR1} + \rho_{NIR}} \tag{4}$$

$$\text{NDWI} = \frac{\rho_G - \rho_{NIR}}{\rho_G + \rho_{NIR}} \tag{5}$$

We chose the above three vegetation indices to analyze the spectral differences between the ancient tombs under the forest and the non-ancient tombs under the forest. Since the Sentinel1 images started in 2019, the vegetation index timing-series feature changes after Savizky–Golay filtering during 2019–2020 are shown in Figures 7–10.

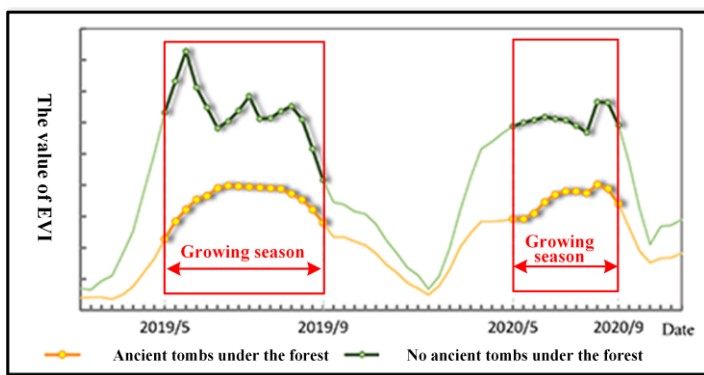

**Figure 7.** Timing-series features of EVI.

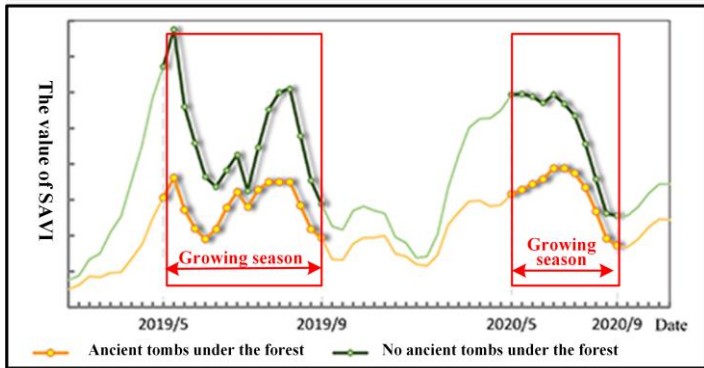

**Figure 8.** Timing-series features of SAVI.

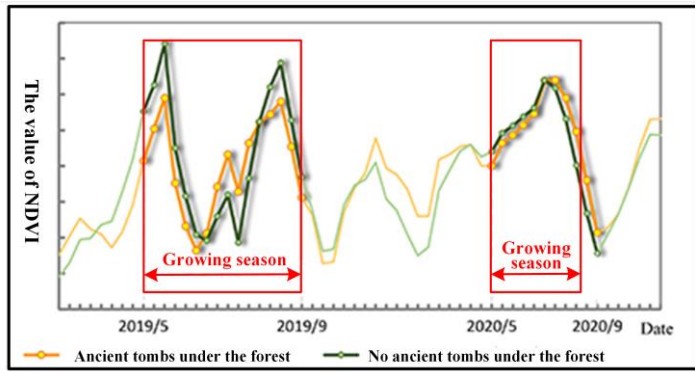

**Figure 9.** Timing-series features of NDVI.

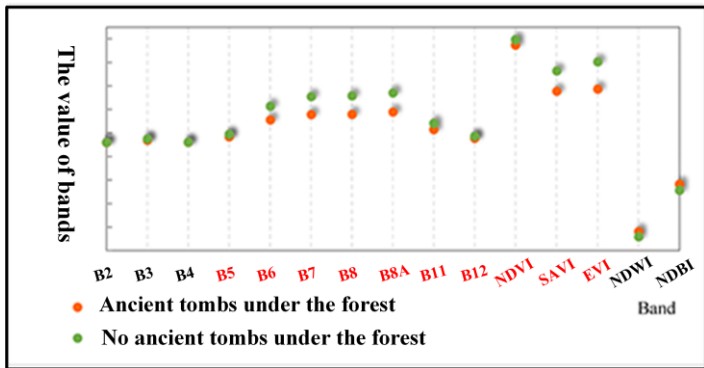

**Figure 10.** Average value of bands in Sentinel-2 in August 2019.

Ancient tombs and non-ancient tombs under the forest exhibit apparent timing-series features in the vegetation index, particularly during the vegetation's growth season, as we can observe from the spectral timing-series features discussed above. Although it is not immediately apparent in NDVI, the improved vegetation index of EVI and SAVI shows the difference. Compared to the forest covered on the surface of the ancient tombs area, the growth tendency of the forest cover on the surface of non-ancient tombs area is noticeably higher. In addition, we select the August image with the most obvious difference on the timing-series curve and observe the values in Sentinel 2 image and vegetation index band. It is discovered that the vegetation index varies, which is compatible with the timing-series properties. Therefore, the image in August 2019 was selected for the research.

(1)　Radar bands. The radar backscattering coefficient of Sentinel-1 is sensitive to the object's dielectric properties. Generally speaking, the rougher the surface of the ground object, the stronger the backscattering, and the brighter the color tone reflected in the image. We examined the timing-series features of the VV band and VH band in Sentinel-1 from 2019 to 2020 to detect ground objects, which heavily relies on analyzing the radar backscattering coefficient features of various ground objects, such as Figures 11 and 12.

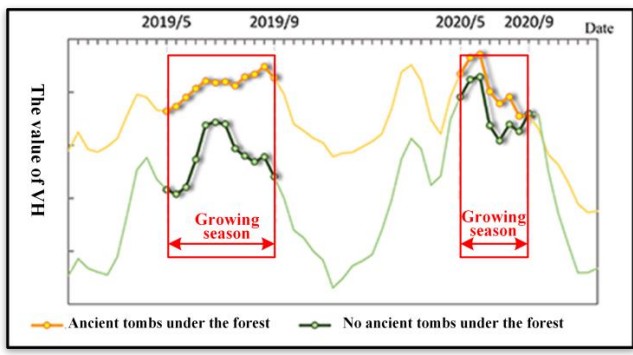

**Figure 11.** Timing-series features of VH.

From the VV and VH timing-series features of Sentinel-1, it can also be seen that there is a relatively noticeable gap between the ancient tombs and non-ancient tombs area under the forest, which reaches its maximum in June and July every year. The VV band spectral value of the non-ancient tombs area is higher than that of the ancient tombs area, but it is the opposite in the VH band. Therefore, the radar bands can also be used to construct features.

We mainly explored the spectral features of the ancient tombs under the forest in the radar bands of Sentinel-1 and the vegetation index and texture bands constructed by Sentinel-2. At the same time, there are differences in the various bands of Sentinel-2 multispectral images. Sentinel-1 has limited use in archaeological research, so we decided to use a combination of two bands, as shown in Table 5.

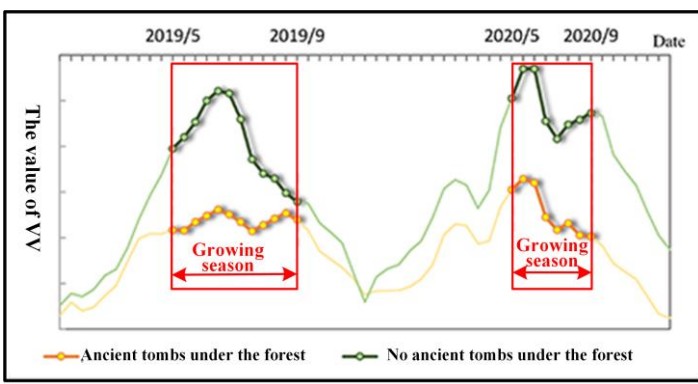

**Figure 12.** Timing-series features of VV.

**Table 5.** Band combinations in this study.

|  | Sentinel-1 | Sentinel-2 |
|---|---|---|
| Combination 1 | - | B2, B3, B4, B5, B6, B7, B8, B8A, B11, B12, NDVI, SAVI, EVI, B8_ASM, B8_CON, B8_CORR, B8_ENT |
| Combination 2 | VV, VH | B2, B3, B4, B5, B6, B7, B8, B8A, B11, B12, NDVI, SAVI, EVI, B8_ASM, B8_CON, B8_CORR, B8_ENT |

### 3.3. Automatic Identification Algorithm Based on Random Forest

Our machine learning technique automatically identifies ancient tombs under the forest uses band combinations based on a random forest classifier. It is based on sample data from ancient tombs under the forest and the chosen spectral properties. Finally, the author performed the necessary accuracy verification of the recognition results. The flow chart of the automatic identification algorithm of ancient tombs under the forest is shown in Figure 13.

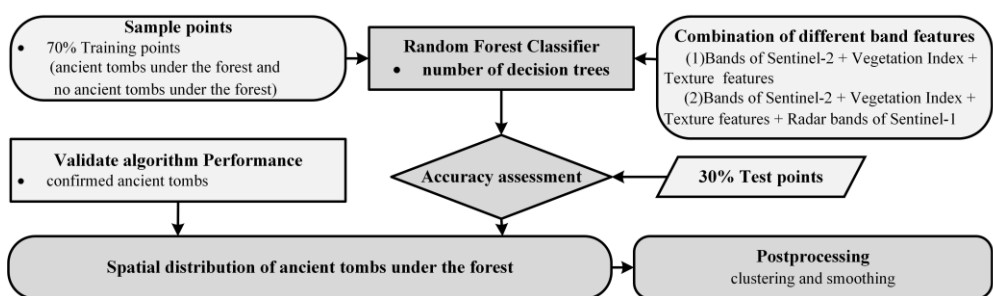

**Figure 13.** Flow chart of automatic identification algorithm of ancient tombs under the forest.

The main algorithm steps of the automatic tombs under the forest based on the GEE platform are as follows, and the pseudo-code is shown in Algorithms 1, specific steps are as follows:

1. Generation of training data. We first import the selected sample data into the GEE. Since the focus of this study is the detection of ancient tombs under the forest, only two categories are selected for the sample data, namely ancient tombs under the forest and non-ancient tombs under the forest, which are represented by "1" and "2". However, to ensure the over-fitting problem of the model, 70% of them are randomly used as training data and the remaining 30% as verification data.

2. Construction of band feature combinations. Following our exploration of the spectral features of ancient tombs beneath the forest, we primarily choose two band feature combinations to detect the influence of different band feature combinations on the

results. There are fifteen bands: two radar bands for Sentinel-1; ten bands for Sentinel-2, except for the B1 and B9 bands; and three vegetation indices. We chose two band combinations, as shown in Table 5. Furthermore, in order to focus on the research object, we masked non-study objects.

3. The training data are used to train the random forest classifier. The random forest classifier mainly uses the bootstrap resampling method to select *n* samples from all the sample data randomly. Each sample has K features, and each sample randomly selects k features (k ≤ K). It sets the best segmentation attribute as the node to establish the optimal decision tree model, combines multiple decision trees for prediction and obtains the optimal classification result through voting. One of the important parameters for random forest classifier detection is the number of decision trees, which determines the number of integrated decision trees. The larger the value, the better the model convergence, but the running time will increase. And when the number of trees is too large, it will be oversaturated. At the beginning of this study, 100 trees were selected to try because some studies have proved that this is the number that can obtain the best results. But in the end, the influence of the different number of decision trees on the experimental accuracy was calculated for this study, and it was found that the classification accuracy is the highest when the number of decision trees is 25.

4. The target classification object will be obtained using the trained random forest classifier to iteratively classify the feature band combination.

5. Independent 30% validation data are used to verify the classification accuracy. We mainly use the confusion matrix, including producer accuracy, user accuracy, overall accuracy (OA), and Kappa coefficient.

$$\text{OA} = \frac{\sum_{i=1}^{n} N_{ii}}{N} \tag{6}$$

$$\text{Kappa} = \frac{N \sum_{i=1}^{n} N_{ii} - \sum_{i=1}^{n} N_{i+} \times N_{+j}}{N^2 - \sum_{i=1}^{n} N_{i+} \times N_{+j}} \tag{7}$$

In the formula, *N* is the total number of samples; *n* is the number of all categories; $N_{i+} = \sum_{j=1}^{n} N_{ij}$ represents the number of samples divided into *i* categories; $N_{+j} = \sum_{i=1}^{n} N_{ij}$ represents the number of samples belonging to category *j*.

6. Ground validation. The overall performance of the automatic detection model is evaluated according to the number of correctly identified ancient tombs that are not involved in the calculation.

7. The spatial distribution of ancient tombs under the forest identified by machine learning is relatively fragmented. Hence, we perform spatial filtering on the specified results to smooth the image and perform spatial connectivity processing to remove small patches.

---

**Algorithms 1.** The implementation process and part of pseudocode

---

1: **Input**: D – Sample data set, A - Band feature set
2: **for** b = 1 to B **do**
3:      Dn=sub_D        #Draw a bootstrap size n from the D.
4:      Am=sub_A        #Randomly select m from A.
5:      splitpoint(Am)    #Pick the best split-point from Am.
6:      node=two_subnode    **return**    #Split the node into two daughter nodes.
7: **end for**
8: **Output**:{T1,T2, . . . ,TB}
9 : $\hat{C}_{rf}^{B}(x) = \frac{1}{B} \sum_{b=1}^{B} T_b(x)$    #Make a Prediction at a new point x toregression.
10 : $\hat{C}_{rf}^{B}(x) = majority\ vote\{\hat{C}_b(x)\}_1^B$    #$\hat{C}_b(x)$ is the class prediction of the bth random-forest tree.

---

## 4. Results

### 4.1. Algorithm Accuracy Verification

The accuracy of the two kinds of band combinations we used in the automatic detection algorithm of tombs under the forest is shown in Table 6.

**Table 6.** Algorithm accuracy verification.

| Class | 1 | 2 | Producer Accuracy | Class | 1 | 2 | Producer Accuracy |
|---|---|---|---|---|---|---|---|
| 1 [1] | 457 | 14 | 96.41% | 1 | 453 | 12 | 97.42% |
| 2 [2] | 4 | 151 | 94.38% | 2 | 4 | 175 | 96.15% |
| User accuracy | 99.13% | 89.88% | | User accuracy | 99.12% | 91.15% | |
| OA: 95.68% Kappa: 90.47% | | | | OA: 96.57% Kappa: 92.97% | | | |

[1] The ancient tomb area under the forest, [2] The non-ancient tomb area under the forest.

From the overall accuracy of the algorithm, band combination 2 is lower than band combination 1. The OA slightly increases from 95.68% to 96.57%, but the Kappa coefficient increases from 90.47% to 92.97%. For the single category, the producer accuracy and user accuracy of the non-ancient tombs area under the forest have improved, indicating that the addition of the radar bands is beneficial for separating the ancient tombs area and non-ancient tombs area under the forest. Due to the great reliability of the detection technique, the two major feature combinations chosen in this work can be used to detect ancient tombs under the forest.

### 4.2. Ground Validation

The confirmed ancient tombs in Baling Mountain must also be used to verify the correctness of the algorithm detection results. Although the study's chosen image was taken in 2019, the confirmed data were obtained in 2010. Therefore, it is necessary to remove all ancient tombs located in non-forest land, and finally, there are 190 ancient tombs covered with forest land. The confirmed data are superimposed with the detection results of ancient tombs under the forest. We found that 163 ancient tomb points were detected under combination 1, while under the band combination 2, 167 ancient tomb points were detected. And the accuracy increased from 85.78% to 87.89%, as shown in Table 7. The detection results are better, and the addition of Sentinel-1 radar bands improves the detection results, but the improvement is limited.

**Table 7.** Ground validation verification.

| | Ancient Tombs Consistent with the Test Results | Measured Number of Ancient Tombs | Accuracy |
|---|---|---|---|
| Combination 1 | 163 | 190 | 85.78% |
| Combination 2 | 167 | 190 | 87.89% |

### 4.3. Spatial Mapping of Automatic Detection in Baling Mountain

Our detection aims to obtain the spatial location of ancient tombs under the forest in the study area. After the algorithm of ancient tombs under the forest is verified, it is applied to detecting ancient tombs in Baling Mountain. Figure 14 shows the target object's spatial distribution map, namely the ancient tomb under the forest, under the combination of two wave band features, which is basically the same as the overall spatial distribution. Most of those in the north are concentrated in blocks, while those in the south are scattered.

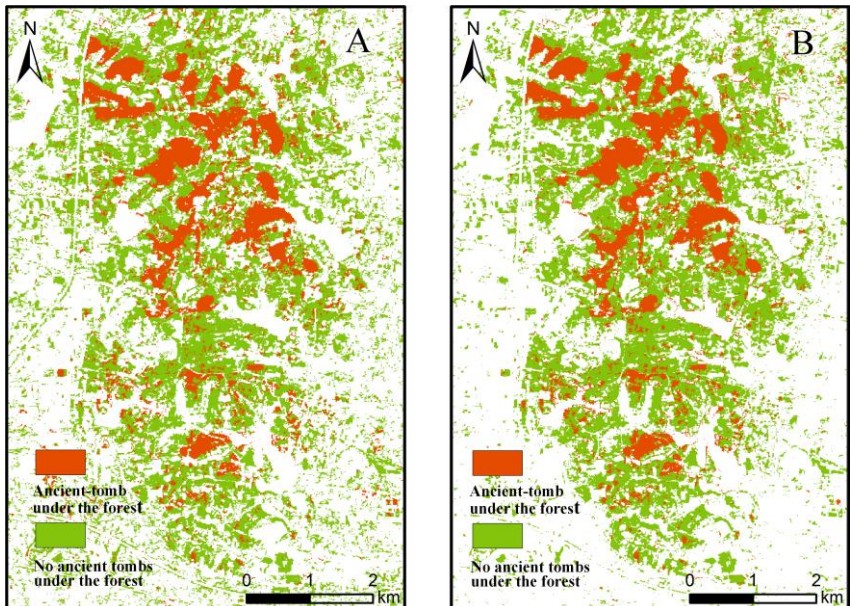

**Figure 14.** Spatial distribution identification of ancient tombs under the forest in Baling Mountain. (**A**) Result of combination 1. (**B**) Result of combination 2. Although the spatial distribution of the two band feature combinations is generally similar, the detection results are improved after adding the radar bands of Sentinel-1, so we carry out the operations, as mentioned earlier, on the spatial recognition results of combination 2 in order to increase algorithm accuracy and ground verification. We overlay the processed results with the known tomb sites, as shown in Figure 15.

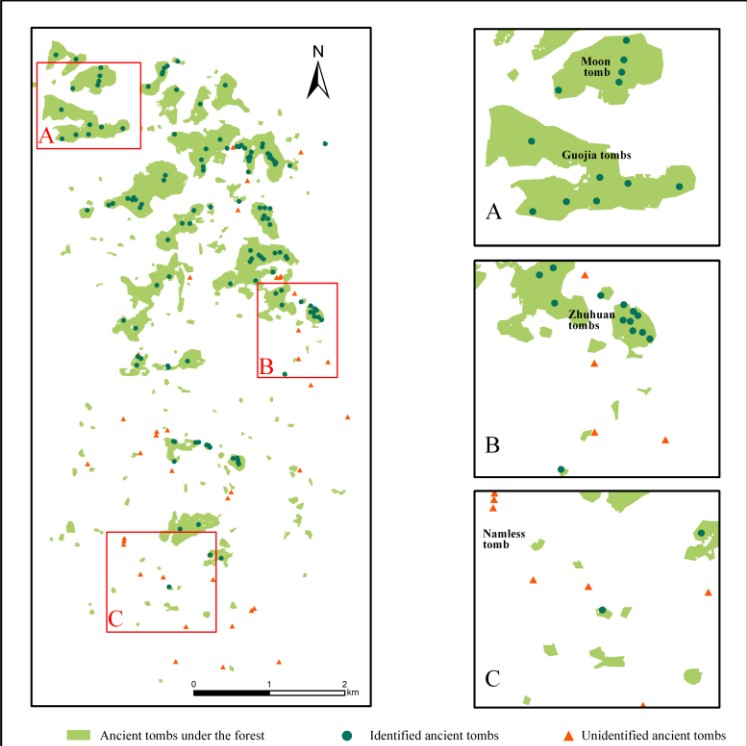

**Figure 15.** The superimposed comparison of the spatial distribution of the ancient tombs under the forest after smoothing and the confirmed tombs. (**A**) It is the area where ancient tombs are large. (**B**) It is the area where ancient tombs are relatively concentrated. (**C**) It is the area where ancient tombs are scattered.

From the comparison between the smoothed distribution of ancient tombs under the forest and the confirmed ancient tombs, A, which is the northwest part of the study area, has the best recognition effect because there are relatively significant large-scale ancient tombs in this area, such as the tombs of "Moon Tomb" and "Guojia Tomb". A is the Figure 15A. The size of the "Moon Tomb" is about 24,073 square meters. B is the central and eastern parts of the study area. The identification effect is better. However, the identification effect of the C is poor, that is, in the southern part of the study area. C is the Figure 15C.

## 5. Discussion

Authors should discuss the results and how they can be interpreted from the perspective of previous studies and of the working hypotheses. The findings and their implications should be discussed in the broadest context possible. Future research directions may also be highlighted.

Comparing the areas with inaccurate identification results with the Sentinel-2 true color image and DSM image, it is found that the areas with the best classification results, such as Figure 16A,B, are mainly massive tombs or tombs clustered together. The areas with poor classification results, such as Figure 16B,C, are mainly discrete and independent ancient tombs. Additionally, the surroundings of ancient tombs are other types of ground objects, such as grassland or buildings. Because the image's resolution in this study is only 10 m, the spectral features of the mixed area of feature types are easy to fuse with the surrounding non-ancient tombs area, resulting in spectral confusion and the disappearance of features [77]. Therefore, the identification of discrete small tombs is not suitable. It is necessary to use higher spatial resolution images or a variety of models and methods, such as linear spectral mixing models, to solve the problem of mixed pixels caused by resolution [78–80]. Currently, the unmixing method mainly uses hyperspectral images [81], but it has been used in many fields [78,82], such as national defense, mining, agriculture, etc. Therefore, it is feasible to use the hyperspectral images unmixing in ancient tombs. There are some identified areas in the distribution of ancient tombs under the forest after identification. The spectral features we are exploring have certain separation effects on them. Still, the spectral features of vegetation are the result of complex effects of various factors, such as vegetation, soil, environment, and water [83]. In the stage of exploring spectral features, in addition to the three vegetation indices we adopted, we also tried other vegetation indices. Still, results are not obvious, so the introduction of image data with more spectral channels is useful to find the difference between the two in quiet places [84] to achieve better recognition results.

Technically speaking, we mainly designed an automatic algorithm for ancient tombs under the forest based on information and combined it with random forest in machine learning. Random forest is suitable for small samples, such as this study, and has the advantages of fast speed, but it has poor generalization ability. Deep learning with good generalization ability is currently lacking in research on medium- and low-resolution remote sensing image classification. Deep learning has shown great potential in the feature representation of remote sensing time series [85,86]. Suppose we want to improve the detection accuracy of small tombs in the Baling Mountains. In this case, we need to combine the constructed features with deep learning methods to find new ways to solve the archaeological problems of ancient tombs in the forest.

The identification method of ancient tombs under the forest shows that it is feasible to detect ancient tombs using medium- and low-resolution images. Still, the success of this study is inseparable from the fine Lidar data. Lidar data provide a huge reference for selecting sample points in the ancient tombs under the forest. However, if the method is extended to other or larger regions, the acquisition of the sample data is essential. Many famous cultural relic areas at home and abroad have performed archaeological work for many years. The combination of archaeological results and GIS to establish a database that is conducive to global spatial analysis is of great significance to the future research and protection of archaeology [87,88], so we can consider calling on these databases to obtain

sample data or seek data support from other regional archaeological departments. Adding more archaeological data will place good research value on ancient tombs buried under the forest in the blank regions of archaeological knowledge.

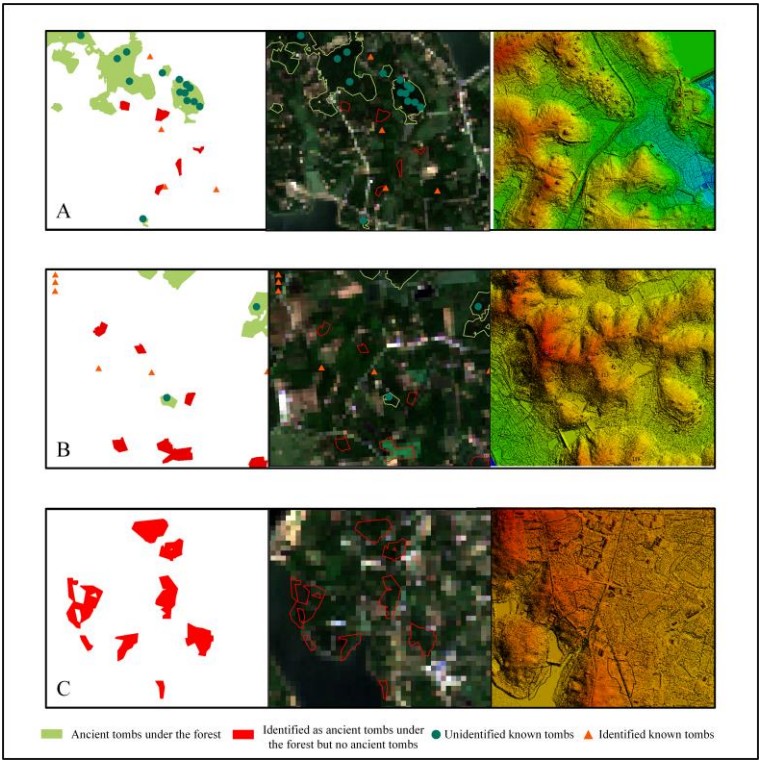

**Figure 16.** Identify areas with errors in results. (**A**) It is the area with better identification results. (**B**) It is the area with the average identification results. (**C**)It is the area with poor identification results.

Time-series indicators can represent the basic biophysical vegetation features well [89]. Furthermore, spectral features should be compared at specific phenological stages to improve the impact of seasonal vegetation changes. In this study area, after thousands of years of the atmospheric–soil–water cycle, the long-term vegetation on the surface of the ancient tombs can be used as a separation feature from the vegetation in the non-ancient tomb areas in time series. However, due to different geographical locations, climate, humidity, and other factors may have different effects on vegetation growth [90–92]. The algorithm parameters are currently only applicable to this study area, and their universality in other regions still needs further exploration.

## 6. Conclusions

This section is not mandatory but can be added to the manuscript if the discussion is unusually long or complex.

Our research focuses on identifying ancient tombs under the forest, and the large number of ancient tombs buried under the ground but not excavated IS crucial for the development and protection of archaeological research. Here, we combined LiDAR and remote sensing images to explore the spectral features of tombs under the forest. We combined spectral features with machine learning to design an automatic identification method of ancient tombs under the forest. After the process is applied to Baling Mountain in Jingzhou, it is found that when compared with the traditional field investigation, this method can directly display the spatial distribution of ancient tombs under the forest on the premise of high feasibility and fast processing speed, so that it has better analysis ability. At the same time, it also provides a certain direction for the field archaeology of archaeologists.

As the work continues, the use of higher-resolution imagery or newer methods may aid in identifying of discrete tombs.

**Author Contributions:** Y.L. carried out the conceptualization, methodology, software, data curation, and writing—original draft preparation. S.W. carried out the visualization and project administration. Q.H. carried out the writing—review and editing. M.A. and P.Z. carried out the investigation. F.Z. carried out the writing—review and editing. All authors reviewed the manuscript. All authors have read and agreed to the published version of the manuscript.

**Funding:** This research was funded by the grants from the National Key R&D Program of China (Grant No. 2020YFC1521903) and the knowledge Innovation Program of Wuhan Basic Research (Grant No.2022010801010431).

**Data Availability Statement:** Our study was run in GEE. So, the data and code can be found here: (https://code.earthengine.google.com/6488ee2bc0b7926a2024d4c3cbb24d6c, accessed on 7 November 2022). The DSM images are not publicly available.

**Acknowledgments:** Thanks to Jingzhou Museum for their help in the field investigation and data collection process of this study.

**Conflicts of Interest:** The authors declare no conflict of interest. The funders had no role in the design of the study; in the collection, analyses, or interpretation of data; in the writing of the manuscript; or in the decision to publish the results.

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
