# Peer review of "Discovering the Ancient Tomb under the Forest Using Machine Learning with Timing-Series Features of Sentinel Images: Taking Baling Mountain in Jingzhou as an Example"

_remotesensing, doi:10.3390/rs15030554_

Round 1
Reviewer 1 Report
This paper introduces an interesting work that combines the LIDAR and multispectral images to discover underground bombs. Overall, it is nice to see that the images can be used to discover the bombs. This paper can be accepted subject to the following questions.
1. Since multispectral images can only capture the surface reflectance of the target, the reviewer wanted to know the depth of the tomb.
2. Some Figures are missing, leading to ‘Error, Reference source not found.
3. Since unmixing was originally used to discover the underground mines, could the unmixing help discover the tomb? It is suggested to provide a discussion following,
doi: 10.1109/TGRS.2021.3074364; doi: 10.1109/TGRS.2021.3081177
Reviewer 2 Report
The introduction appears well structured and extensively describes previous research experiences, the types of data used, their usefulness and the archaeological context of application.
The research design is in line with the required topic and the methods are also widely described in their various stages.
As regards the results, they appear very significant and promising also for research activities in other similar contexts where high vegetation conceals the archaeological remains of tombs.
The research results are presented in a clear way and accompanied by understandable images, they also appear well summarized in the conclusions.
However, reading the text as a whole, some passages appear redundant, as well as some grammatical and punctuation errors that need to be corrected.
Overall, the opinion on the paper is positive and its publication is recommended on the condition of some small changes to the English text.
Author Response
Dear Review:
Thank you for your careful reviewing concerning our manuscript entitled “Discover the ancient tomb under the forest using machine learning with timing-series features of Sentinel imag-es——Taking Baling Mountain in Jingzhou as an example” (ID: remotesensing-2048806). Your comments are all valuable and very helpful for revising and improving our paper, as well as the important guiding significance to our researches.
Point 1: reading the text as a whole, some passages appear redundant, as well as some grammatical and punctuation errors that need to be corrected.
Response 1: Regarding the grammatical and punctuation errors you mentioned, I have asked my English majors to revise. And I also reorganize the full text.
With kindest regards,
Yours Sincerely
Yichuan Liu
School of Remote Sensing and Information Engineering
Wuhan University, Wuhan, China, 430079
Reviewer 3 Report
Dear Authors,
I would like to offer my compliments as I really enjoyed this article.
You have produced a very interesting text to read, which may provide inspiration to other researchers for other analyses, since you use data made available for free.
I have some notes to make, as the text seems to be a bit rushed, especially in the writing and the search for references:
1. please rewrite the abstract avoiding numbers and paragraphs. Make the text more discursive
2. in the keywords insert at least the word satellite or sentinel-2
3. in the title there are two signs that I do not understand
4. ll.30-31 "utensils and articles" -> artefacts and finds
5. ll. 35-42 add reference bibliography
6. ll. 49-52 add reference bibliography
7. ll. 80 Dissolve the acronym DSM (it also appears in the abstract and is not dissolved). Please review and add terminology and LiDAR references on the latest studies and trends in archaeology in the lines above:
e.g.
(i) Chase, A.F.; Chase, D.Z.; Weishampel, J.F.; Drake, J.B.; Shrestha, R.L.; Slatton, K.C.; Awe, J.J.; Carter, W.E. Airborne LiDAR, Archaeology, and the Ancient Maya Landscape at Caracol, Belize. J. Archaeol. Sci. 2011, 38, 387–398.
(ii) Chase, A.; Chase, D.; Chase, A. Ethics, New Colonialism, and Lidar Data: A Decade of Lidar in Maya Archaeology. J. Comput. Appl. Archaeol. 2020, 3, 51–62.
(iii) Lozić, E.; Štular, B. Documentation of Archaeology-Specific Workflow for Airborne LiDAR Data Processing. Geosciences 2021, 11, 26.
(iv) Masini, N.; Abate, N.; Gizzi, F.T.; Vitale, V.; Minervino Amodio, A.; Sileo, M.; Biscione, M.; Lasaponara, R.; Bentivenga, M.; Cavalcante, F. UAV LiDAR Based Approach for the Detection and Interpretation of Archaeological Micro Topography under Canopy—The Rediscovery of Perticara (Basilicata, Italy). Remote Sens. 2022, 14, 6074.
(v) Štular, B.; Lozić, E.; Eichert, S. Airborne LiDAR-Derived Digital Elevation Model for Archaeology. Remote Sens. 2021, 13, 1855.
(vi) Guth, P.L.; Van Niekerk, A.; Grohmann, C.H.; Muller, J.-P.; Hawker, L.; Florinsky, I.V.; Gesch, D.; Reuter, H.I.; Herrera-Cruz, V.; Riazanoff, S.; et al. Digital Elevation Models: Terminology and Definitions. Remote Sens. 2021, 13, 3581.
8. l. 161 'archaeology.The random' lacks space
9. ll. 165-175 Completely to be revised. References missing. In recent years dozens of texts (also on MDPI) have been published on the use of GEE for Cultural Heritage, please add references and explain better.
10. par. 3.1.1. I did not understand whether you are using a DSM or a DTM/DFM. There is a difference between the models, you will find it described in the bibliography listed above. Please specify better. Also add which version of Global Mapper you are using.
11. ll. 243-244 ... something went wrong!
12. 3.2.2. Why does it start with a kind of paragraph?
13. Please add studies (references) in which vegetation indices and Sentinel-1 and 2 are used for archaeology. Many have been done. Many studies have also been done on the best time to grow and study crop-marks. e.g.
(i) Kaimaris, D.; Patias, P.; Tsakiri, M. Best period for high spatial resolution satellite images for the detection of marks of buried structures. Egypt. J. Remote Sens. Space Sci. 2012, 15, 9–18.
(ii) De Guio, A. Cropping for a Better Future, Vegetation Indices in Archaeology. In Detecting and Understanding Historic Landscapes; PCA Studies; SAP: Mantova, Italy, 2015; pp. 23–60. ISBN 978-88-87115-99-4.
13. l. 310 ... something has gone wrong!
14. l. 352 ... something has gone wrong!
15. Is it possible to have a land cover map? To understand the types of forest and vegetation?
16. There are typing errors in the conclusions.
Reviewer 4 Report
1) the paper lacks a introduction to the object of study. Speaking about "ancient tombs" is vague, and the historical background given is vague as well. The authors must specify which kind of tomb (large, small, which plan, and so on) refers to each historical period considered, since the tombs can span some 2000 years (from Warring states to Ming) and extract results coherently.
2) the authors miss completely and should refer to the existing literature on satellite imagery analisiys of ancient Chinese tombs and (related) Japanese and Egyptian tombs, published in recent years by the Milan Politecnico group and by others, see for instance:
Magli, G Royal mausoleums of the western Han and of the Song Chinese dynasties: A satellite imagery analysis Archaeological Research in Asia, Volume 15, September 2018, Pages 45-54
Parcak, S. (2015). Archaeological looting in Egypt: A geospatial view (case studies from Saqqara, Lisht, and el Hibeh). Near Eastern Archaeology, 78(3), 196-203.
3) On-field recognition of features identified on data should be referred, at least at the sample level
4) The text contains trivial errors: the suggestions for each chapter present in the template, reference errors, and so on
After these changes, I believe the paper can be accepted
